# An On-Orbit Task-Offloading Strategy Based on Satellite Edge Computing

**DOI:** 10.3390/s23094271

**Published:** 2023-04-25

**Authors:** Yifei Hu, Wenbin Gong

**Affiliations:** 1Innovation Academy for Microsatellites, Chinese Academy of Sciences, Shanghai 201210, China; huyf2021sh@163.com; 2University of Chinese Academy of Sciences, Beijing 100049, China

**Keywords:** MEC, LEO satellites, collaborative computing, task scheduling, satellite node constraints

## Abstract

Satellite edge computing has attracted the attention of many scholars due to its extensive coverage and low delay. Satellite edge computing research remains focused on on-orbit task scheduling. However, existing research has not considered the situation where heavily loaded satellites cannot participate in offloading. To solve this problem, this study first models the task scheduling of dynamic satellite networks as a minimization problem that considers both the weighted delay and energy consumption. In addition, a hybrid genetic binary particle swarm optimization (GABPSO) algorithm is proposed to solve this optimization problem. The simulation results demonstrate that the proposed method outperforms the other three baseline algorithms.

## 1. Introduction

In recent years, the terrestrial Internet has rapidly developed, with applications such as smart cities and environmental monitoring [1,2] attracting widespread attention. However, terrestrial Internet services are concentrated mainly in urban areas, and providing quality services in remote areas, such as islands, oceans, and deserts, is challenging. The instabilities of terrestrial networks become apparent when faced with natural disasters, such as floods and earthquakes [3]. Satellite networks have significant advantages, such as wide global coverage and high destructive resistance. They have been used in emergency communications, navigation, positioning, and smart city applications [4,5], effectively compensating for the lack of terrestrial Internet services and providing critical support for 6G global interconnection [6,7]. However, previous research has considered primarily satellite networks as relay networks in a satellite–ground fusion architecture. Ground access terminal services and raw satellite remote sensing data, for example, are transmitted back to the ground cloud center for unified processing, despite the possibility of performing tasks directly on the satellite [3,8,9]. Both result in significant delays and the waste of valuable satellite communication resources [10,11].

Mobile edge computing (MEC) [12] is an emerging architecture that brings the traditional cloud-centric computing model down to the edge of the user node. It provides the services and computing power needed at the user’s periphery, creating service edge nodes with low latency and high processing rate for a better quality of service (QoS). MEC principles and low-latency, high-capacity low-Earth orbit (LEO) satellite networks are combined to create edge computing satellites (ECS). In this manner, satellite acquisition data can be processed in real-time at satellite edge nodes [4,8,9,13], conserving bandwidth and allowing for quick mission reaction. The TIANSUAN Constellation test satellite [14], co-chaired by Beijing University of Posts and Telecommunications and other institutions, for example, has validated remote sensing image inference and computing services on orbit with the equipped KubeEdge and Sedna edge intelligence inference platform. The researchers also compared it with traditional ground-based backhaul analysis strategies and verified that on-orbit edge processing can effectively reduce transmission traffic and latency. At the same time, they also investigated the on-orbit verification of the distributed cognitive service-oriented architecture for the 6G core network.

However, satellite on-orbit processing tasks face the challenge of a single satellite’s limited computing power, resulting in delayed processing of tasks and difficulties in meeting user requirements due to long task delays. Additionally, reducing energy consumption in satellite edge computing is a major concern for scholars [15,16]. Therefore, utilizing multiple satellites for assisted computing is a valuable research direction to achieve smaller latency and energy consumption.

The contribution of this research work includes the following: This research proposes a computation offloading strategy based on satellite edge computing, where large dependent tasks can be divided into multiple subtasks and computed on other satellite nodes.On the basis of the shortcomings of computation offloading strategies for satellite edge computing of other literatures, study paper considers the scenarios where the satellite network topology changes, and high-load satellites are not involved in offloading. The optimization problem of weighted minimization of task completion delay and energy consumption is then investigated.To solve the optimization problem, the modified GABPSO algorithm is utilized. The proposed algorithm is extensively simulated, demonstrating its superior performance compared with other benchmark algorithms.

The remainder of this paper is organized as follows: Section 2 summarizes the related work. Section 3 describes the system model and problem formulation. Section 4 describes the proposed hybrid genetic binary particle swarm algorithm and analyzes the simulation results. Section 5 concludes the paper.

## 2. Related Works

Task scheduling in edge computing can be divided into two categories: static task scheduling and dynamic task scheduling [17]. When task information and network information are known, static task scheduling can be used directly for task offload scheduling.

Dynamic task scheduling involves reassigning the scheduling policy at each scheduling moment when the number of tasks, network information, and other factors change at any time. For the purpose of this paper, we focus on static task scheduling.

Static tasks that are offloaded to the edge can be classified into two types: independent tasks and dependent tasks [18]. Independent tasks can be split into multiple tasks processed in parallel, and each node returns the result after completing the task processing. In contrast, dependent tasks include several subtasks with logical dependencies. In addition, the processing of a subtask can be performed when all the preceding subtasks of the subtask are completed. 

Due to the constraint relationship among subtasks, scheduling dependent tasks is more challenging than scheduling independent ones. With the rise of big data, dependent tasks are becoming more common, including target tracking and identification, which require combining and processing multiple tasks [18]. As a result, developing effective scheduling strategies for dependent tasks is crucial. Although static task scheduling in terrestrial MEC scenarios has been extensively studied, research on task scheduling for LEO satellite constellations is still in its early stages, particularly for dependent tasks.

For independent-type task scheduling, Ren et al. [19] proposed an inter-satellite collaborative computation method for formation-flying satellites. The authors characterized the formation-flying satellite network using a weighted undirected graph, dividing the computational tasks into multiple parallel computational subtasks assigned to each satellite node and solving the delay optimization problem under the energy consumption constraint using the modified particle swarm algorithm (MPSO). However, because the work was limited to formation-flying satellites with a constant topology, its applicability for task scheduling of LEO satellite constellations with dynamic topologies is limited.

Chen Wang [11] presented a strategy for LEO satellite collaborative computing. The authors used time-expanded graphs to generate a steady-state matrix for dynamic LEO satellite networks. Furthermore, a generalized discrete algorithm based on transmission capacity and computational power addressed the time-delay optimization problem for multi-satellite collaborative computing applications. The scheduling of independent tasks was simpler because there were no logical dependencies between tasks.

For dependent task scheduling, Wu et al. [20] proposed a task collaborative scheduling algorithm for small satellite cluster networks. The research authors assigned dozens of jobs with logical relationships to separate satellites for collaborative computation. Considering that satellite nodes fail, the authors proposed an improved task scheduling strategy based on three heuristic algorithms so that the system can effectively guarantee that all tasks are completed by the deadline as much as possible while also providing some robustness to the scheduling algorithm. The effectiveness of the proposed algorithms was verified by comparing them with genetic algorithms, for example. However, again, the authors considered the same network of small satellite clusters and formation-flying satellites, and the topology between satellites remained fixed, which lacked a reliable reference value for the dynamic LEO satellite network.

Guo et al. [21] first characterized the LEO satellite network using the weighted time extended graph (WTEG) model, in which a uniform delay weight parameter was added to each edge in the steady-state graph to analyze the delay of the on-orbit computation and transmission. A directed acyclic graph (DAG) was used to characterize the task model, and the nodes and edges of the task model were mapped to the steady-state graph to find the minimum task completion delay. The authors employed a binary particle swarm algorithm to optimally solve the optimal mapping problem and verify the algorithm’s feasibility in comparison with ground cloud processing and other basic scheduling algorithms.

Han et al. [4] constructed a satellite edge cluster computing architecture using LEO and geostationary earth orbit (GEO) satellites as edge nodes for collaborative task computing and characterized the logical relationships and constraints among subtasks using the DAG model. Furthermore, the author designed a scheduling algorithm that considered the dynamic changes in the priority and link bandwidth of subtasks in different time slots. At each scheduling moment, the unresolved subtasks were assigned to the appropriate satellite nodes for processing to ensure that the corresponding metrics of interest were optimized.

Most authors included all edge computing satellites in the spectrum of task scheduling assignable nodes in the work mentioned above. In fact, due to factors such as unequal population distribution and varying business demands, the load of each satellite node varies significantly. Enough computing power for new task processing is difficult for high-load satellites. However, this problem has not been considered in any the above research. 

At the same time, the satellite was powered by solar energy, and the energy consumption was an optimization target of great interest in satellite terrestrial networks [22,23]. This was basically not mentioned by the authors in the above work. As a result, the remainder of this paper will focus on investigating a strategy in which satellite nodes with high loads are excluded from task scheduling in the dynamic LEO satellite network task scheduler. Task latency and energy consumption will be considered together. In addition, the research concentrates on the dependent tasks. The difference between our work and the existing literature is summarized in Table 1.

## 3. Model Introduction and Problem Analysis

### 3.1. Satellite Edge Computing Architecture

In this paper, we introduce a classic satellite edge computing architecture, as shown in Figure 1. Task requests uploaded by users on the ground can be continually received by the LEO satellite during its motion. These tasks include the analysis of monitoring data from some sensors, assistance with communication from ocean-going ships, requests for emergency communication from the ground, and analysis of remote sensing images from the satellite itself. Traditionally, the satellite acts as a relay node to transmit the tasks back to the ground central station, i.e., the ground cloud center, for batch processing. However, as the satellite gains access to more powerful computing resources, it may think about processing tasks on the satellites while in orbit instead of sending them back to the ground cloud center. The satellites will carry the edge computing servers. The functional components are shown in Figure 1. The satellite edge computing server can autonomously perform work, such as resource allocation and task scheduling, in orbit. This edge computing satellite model, which is close to the users on the ground, may efficiently decrease the backhaul network traffic while also reducing task-processing latency.

When a satellite receives many tasks, it considers the resource usage of each satellite in the constellation. In addition, a suitable strategy for offloading in orbit is developed. The inter-satellite offloading issue for a single dependent task in this edge computing satellite scenario is the focus of this article. Each satellite can establish contact with the four satellites around it using the inter-satellite link (ISL), as depicted in Figure 2. Due to the various operation conditions, each satellite has a unique load state, which can be broadcast to other satellites via the ISL. For example, when SAT1 receives a complete task request, it will select the appropriate low-load satellite in the constellation to offload parts of subtasks, while the high-load satellite cannot be selected for offloading.

### 3.2. SatTEG Construction

Unlike traditional terrestrial MEC networks, which have a fixed topology, satellite MEC networks have periodic topological changes due to the high-speed movement of satellites. It makes the transmission time delay between satellites uncertain. The fundamental difficulty to be addressed in the subsequent study is how to design the satellite MEC network as a mathematical model in a reasonable manner. Like the previous works [11,24,25], our paper first characterizes the dynamic satellite network using time-expanded graphs. The LEO satellite network experiences periodic topological changes as it travels around the globe, separating each operational period T into N time slots and determining the length of each time slot $\Delta t=\frac{T}{N}$. Within each time slot, the topological state of the satellite network can be considered relatively stable and constant. 

The set of all satellites in the LEO satellite network can be stated as $V=\left\{v_{1},\cdots ,v_{i},\cdots ,v_{d}\right\}$, in which d is the number of LEO satellites. In each time slot, every satellite establishes a connection with the two satellites preceding and following it in the same orbit, as well as the two satellites closest to it in adjacent orbits. The resulting network of connections among satellites in time slot t can be represented by the connection status C(t).
(1)$$C\left(t\right)=\left(\begin{array}{ccc} C_{11} & \cdots & C_{1d}\\ \vdots & \ddots & \vdots \\ C_{d1} & \cdots & C_{dd} \end{array}\right)$$
where $C_{\mathrm{ij}}=1,i,j\in d$ means that the two satellites are connected, otherwise $C_{\mathrm{ij}}=0$.

Further, the connectivity of LEO satellite nodes during their operation cycle can be expressed as a route table SatTEG by combining the connection status $C\left(t\right)$ of all time slots. By using SatTEG, we can obtain the connectivity path of any satellite in any time slot during the satellite network operation. In addition, we can get the number of hops transmitted between satellites, denoted as hop.

### 3.3. Task Model

This section discusses a dependent task model. We assume that in the satellite working stage, a satellite node can achieve a remote sensing image online reasoning task [14]. The image reasoning task can be divided into several dependent subtasks. Each subtask can be assigned to different satellites for AI online analysis according to the satellite channel conditions and computing power. The division method for dependent subtasks [26] and AI on-orbit analysis [27,28] have been investigated to some extent, but they are not the topic of this paper. Hence, they are not discussed in detail.

As shown in Figure 3, we use a typical DAG to represent the subsequent research’s subtask dependence model. The DAG can be expressed as $\varphi =\left(O,E\right)$. $O=\left\{o_{1},\cdots ,o_{p}\right\}$ represents p subtask nodes, and $E=\{e_{ij}\left|\left(i,j\right)\in p\right\}$ denotes the set of directed edges in DAG. Furthermore, we define any subtask $o_{i}\left(i\in p\right)$ as $o_{i}=\left\{D_{i},\zeta_{i}\right\}$. $D_{i}\left(Mb\right)$ and $\zeta_{i}\left(CPUcycle/Mb\right)$ indicate the quantity of computation and computational complexity of subtasks, respectively. When the subtask $o_{i}$ calculation is finished, the calculated result data quantity $RDo_{i}$ must be transferred to the subtask $o_{j}$, as indicated by the weighted directed edge $e_{ij}=RDo_{i}$.

At the same time, we represent the precursor subtasks set of the subtask $o_{i}$ as $PRE_{i}$. The subtask $o_{i}$ is authorized to begin computation when all the subtasks in $PRE_{i}$ have been completed and the appropriate calculation results have been successfully sent to the satellite node where the subtask $o_{i}$ is situated.

According to the above analysis, the task scheduling strategy in the LEO satellite network is the mapping scheme from task graph DAG to the SatTEG.

### 3.4. Mapping Analysis

#### 3.4.1. Node Mapping

We define $m\left(o_{i},v_{j}^{n}\right)=1$ to mean that the subtask $o_{i}$ is assigned to the nth time slot jth satellite node for computational processing. When $m\left(o_{i},v_{j}^{n}\right)=0$, it indicates that the variable is not allocated. For all subtasks and satellite nodes of the complete time slot, the mapping connection can then be written as a decision matrix M
(2)$$\begin{array}{c} M=\left[\begin{array}{ccccc} m\left(o_{1},v_{1}^{1}\right) & \cdots & m\left(o_{1},v_{d}^{1}\right) & \cdots & m\left(o_{1},v_{d}^{N}\right)\\ m\left(o_{2},v_{1}^{1}\right) & \cdots & m\left(o_{2},v_{d}^{1}\right) & \cdots & m\left(o_{2},v_{d}^{N}\right)\\ \vdots & \vdots & \ddots & \vdots & \vdots \\ m\left(o_{p-1},v_{1}^{1}\right) & \cdots & m\left(o_{p-1},v_{d}^{1}\right) & \cdots & m\left(o_{p-1},v_{d}^{N}\right)\\ m\left(o_{p},v_{1}^{1}\right) & \cdots & m\left(o_{p},v_{d}^{1}\right) & \cdots & m\left(o_{p},v_{d}^{N}\right) \end{array}\right]. \end{array}$$

Any node $m\left(o_{i},v_{j}^{n}\right)\in M$ needs to satisfy
(3)$$m\left(o_{i},v_{j}^{n}\right)\in \left\{0,1\right\},\forall o_{i}\in O,\forall v_{j}^{n}\in V,$$
(4)$$\sum_{n=0}^{N}\sum_{j=0}^{d}m\left(o_{i},v_{j}^{n}\right)=1,\forall o_{i}\in O.$$

#### 3.4.2. Path Mapping

After each subtask node is completely mapped to the corresponding satellite node, the weighted directed edge $e_{ij}$ between subtasks can be converted to the shortest path, $Path_{end\left(v\left(o_{i}\right)\right)start\left(v\left(o_{j}\right)\right)}$, between mapping nodes. $m\left(o_{i},v_{j}^{n}\right)=1$ guarantees the node assignment of subtasks, and each subtask may span multiple time slots from the start of transmission to the completion of the computation. $start\left(v\left(o_{j}\right)\right)$ signifies the satellite node assigned by subtask $o_{j}$ in the beginning time slot, and $end\left(v\left(o_{i}\right)\right)$ denotes the satellite node in the time slot when the subtask $o_{i}$ computation is completed. The shortest path Path is obtained by Dijkstra’s shortest path algorithm with the route table SatTEG as input. Similarly, we can also obtain the minimum hop, $Hop_{end\left(v\left(o_{i}\right)\right)start\left(v\left(o_{j}\right)\right)}$.

### 3.5. Objective Function

As analyzed in the task model, the subtask $o_{i}$ is authorized to begin computation when all the subtasks in $PRE_{i}$ have been completed. Therefore, the subtasks are not all offloaded in the first time slot. Considering that the subtask assignment may select a certain satellite node after several time slots, we use $T_{wait}^{i}$ to signify the inter-slot duration of waiting before the transmission of this subtask. 

Assume that the source node initiating the scheduling is $v_{1}^{1}$. When subtask $o_{i}$ is assigned from source node $v_{1}^{1}$ to node $v_{k}^{n}$, it needs to wait for n − 1 time slots at the source node. $T_{wait}^{i}$ can be expressed as
(5)$$T_{wait}^{i}=\left(n-1\right)\Delta t.$$

In addition, the original data of subtask $o_{i}$ is transferred from source node $v_{1}^{1}$ to $v_{k}^{n}$ in the following time
(6)$$T_{\mathrm{trans}}^{i}=\frac{D_{i}}{B}\mathrm{ho}p_{i},$$
where $B\left(Mb/s\right)$ denotes the transmission rate of the ISL, and $hop_{i}$ denotes the minimum number of hops required for a subtask $o_{i}$ to transmit to the destination node $v_{k}^{n}$.

Additionally, the energy consumption resulting from the transmission of the subtask $o_{i}$ through the ISL is defined as
(7)$$E_{\mathrm{trans}}^{i}=T_{\mathrm{trans}}^{i}P_{\mathrm{trans}\;},$$
where $P_{trans}$ is the transmission power of the ISL.

When the data of subtask $o_{i}$ are all transmitted to the destination node $v_{k}^{n}$, the computation time for this subtask is
(8)$$T_{\mathrm{com}}^{i}=\frac{D_{i}\zeta_{i}}{C_{k}},$$
where $C_{k}\left(CPUcycle/s\right)$ is the on-orbit processing performance of satellite node $v_{k}$, and the calculated energy consumption of $o_{i}$ on satellite node $v_{k}$ is defined as
(9)$$E_{\mathrm{com}}^{i}=T_{\mathrm{com}}^{i}P_{\mathrm{com}}^{k},$$
where $P_{com}^{k}$ is the computational power of satellite node $v_{k}.$

The processing result RD must be transferred to the node allocated to the succeeding task $o_{j}$ once the subtask $o_{i}$ is computed, and the transmission time is stated as
(10)$$T_{\mathrm{re}}^{i}=\frac{{\mathrm{RD}}_{o_{i}}}{B}\mathrm{ho}p_{i}^{\mathrm{re}},$$
where $\mathrm{ho}p_{i}^{\mathrm{re}}$ denotes the minimum number of hops required for $RDo_{i}$ to transmit to the node allocated $o_{j}$.

The transmission energy consumption of $RDo_{i}$ is defined as
(11)$$E_{\mathrm{re}}^{i}=T_{\mathrm{re}}^{i}P_{\mathrm{trans}}$$

Then, the final completion time of subtask $o_{i}$ is
(12)$$T_{\mathrm{end}}^{i}=T_{\mathrm{wait}}^{i}+T_{\mathrm{trans}}^{i}+T_{\mathrm{com}}^{i}+T_{\mathrm{re}}^{i}.$$
and the final energy consumption of subtask $o_{i}$ is
(13)$$E^{i}=E_{\mathrm{trans}}^{i}+E_{\mathrm{com}}^{i}+E_{\mathrm{re}}^{i}.$$

We have specified that the start of subtask $o_{j}$’s computation must occur after the completion of its preceding subtask set $PRE_{j}$. That is, the original data transfer completion time of subtask $o_{j}$ follows the constraint
(14)$$T_{\mathrm{wait}}^{i}+T_{\mathrm{trans}}^{j}\geq T_{\mathrm{end}}^{\mathrm{PR}E_{j}}.$$

The depth-first algorithm can be used to determine the order in which subtasks are executed. We must perform the calculations in the logical order of the activities for dependent subtasks. The task’s total completion time is then equal to the time it took to complete the last exit subtask $o_{l},$ i.e.,
(15)$$T=T_{\mathrm{end}}^{l}.$$

To simplify the model, the satellite nodes assigned to the exit subtask $o_{l}$ in this study can be thought of as directly sending the calculation results to the ground cloud center after completing the task computation; the accompanying feedback delay is ignored, then $T_{re}^{l}=0$.

The total energy consumption can be defined as
(16)$$\begin{array}{c} E=\sum_{i=0}^{p}E^{i}. \end{array}$$

Furthermore, we define Ω as the set of satellite nodes with high service load, any satellite node $v_{g}\in \Omega$ can only be utilized as an auxiliary node for subtask transmission, and no subtasks can be scheduled for computational processing. The scheduling procedure should then satisfy
(17)$$m\left(o_{i},v_{g}^{n}\right)=0,\forall o_{i}\in O,\forall v_{g}\in \Omega ,\forall n\in N.$$

Both task completion delay and system energy consumption are issues to consider during the satellite task scheduling process. The system cost obtained by weighting them together is defined as
(18)COST=αT+βE.
where α and β are used as weights to indicate the importance given to latency and energy consumption, respectively. In summary, the optimization problem for dependent task scheduling based on SatTEG can be represented as follows:(19)$$\begin{array}{c} min\;COST\\ s.t.\;(2)\left(3\right)\left(4\right)\left(14\right)\left(17\right). \end{array}$$

## 4. Algorithm Introduction and Simulation Analysis

### 4.1. Algorithm Introduction

The binary particle swarm optimization (BPSO) algorithm was utilized to solve the similar model [21]. The BPSO algorithm has a memory function and can converge to a stable solution in a short time, but it is easy for it to fall into a local optimum. The genetic algorithm (GA) algorithm has a wide range of spatial search capability and variational capability, with strong global search capability, and can effectively overcome the problem of falling into a local optimum in the search process, making it suitable for massively parallel computing. To compensate for the limitations of the BPSO algorithm, we combine the BPSO algorithm and the GA in this work to obtain the GABPSO algorithm. It not only ensures a better information exchange mechanism but also avoids the deficiency of falling into a local optimum, enhances the search velocity, and improves the success rate of optimal solutions.

First, we design the position and velocity of the u<U particles in the i<I iteration of the BPSO algorithm as
(20)$$X_{u}^{i}=\left(x_{u}^{i}\left(1\right),\cdots ,x_{u}^{i}\left(k\right),\cdots ,x_{u}^{i}\left(p\right)\right),$$
(21)$$V_{u}^{i}=\left(v_{u}^{i}\left(1\right),\cdots ,v_{u}^{i}\left(k\right),\cdots ,v_{u}^{i}\left(p\right)\right).$$

In (23), $x_{u}^{i}\left(k\right)\in X_{u}^{i}≜\left[1,J\right]\left\{0,1\right\}$, where $\left[1,J\right]$ represents a 1×J-dimensional array, and $2^{J}>d\times N$. Then each particle position $X_{u}^{i}$ is represented by a binary combination of p groups, and the corresponding task allocation node can be obtained by combining the decision matrix M after decimal decoding. That is, $\forall X_{u}^{i}\left(u\in U,i\in I\right)$ is a possible solution to the objective function. In (24), the initial value of $v_{u}^{i}\left(k\right)\in V_{u}^{i}$ is defined as a random array within [0,1], which is matched with $X_{u}^{i}$.

The velocity update formula is
(22)$$V=W\times V+C1\times r1\times \left(\mathrm{pbest}-X\right)+C2\times r2\times \left(\mathrm{gbest}-X\right),$$
where W is the inertia weight, r1,r2 is a random number between 0 and 1, and C1,C2 is the learning factor.

The position update formula is
(23)$$x_{u}^{i+1}\left(k\right)\left[j\right]=\left\{\begin{array}{cc} 0, & r\geq \frac{1}{1+e^{-v_{u}^{i+1}\left(k\right)\left[j\right]}}\\ 1, & r<\frac{1}{1+e^{-v_{u}^{i+1}\left(k\right)\left[j\right]}} \end{array}\right.$$
where r is a random number.

The GA’s crossover and mutation operations are introduced after updating particle positions. The updated particle positions in the ith iteration is polled in turn. A particle position $X_{f}^{i}$ is randomly chosen from the particle population, and its partial encoding is crossed with the polled particle position. The crossed particle position is inverted with a particular probability to obtain the mutation. Finally, the particle positions are utilized as input of the decision matrix M to find the fitness function, which is the value of the optimization objective function COST in (19). When we find the minimum fitness function, the minimum system cost can be obtained.

The specific steps of the improved GABPSO hybrid algorithm are shown in Algorithm 1.
**Algorithm 1** GABPSO Task Scheduling AlgorithmInput: task φ; route table SatTEG; high load satellites set Ω; weights α,β
; ISL bandwidth B
$;\;\mathrm{power}\;P_{trans},P_{com}$
1: Initialization a particle swarm2: 
while u<Udo
3:   
$\mathrm{Initialization}\;X_{u}^{0},V_{u}^{0}$
4: end while5: Calculate fitness function fit of the particle swarm by substituting the particles into decision matrix M, i.e., fit=COST.6: 
$\mathrm{Set}\;\mathrm{the}\;\mathrm{current}\;\mathrm{position}\;\mathrm{as}\;\mathrm{the}\;\mathrm{best}\;\mathrm{position}\;\mathrm{for}\;\mathrm{each}\;\mathrm{particle}\;P_{xbest}$
7: 
Set the position of the particle with the smallest fit
$\;\mathrm{among}\;\mathrm{all}\;\mathrm{particles}\;\mathrm{as}\;\mathrm{the}\;\mathrm{global}\;\mathrm{best}\;\mathrm{position}\;G_{xbest}$
8:  for i<I do9:  for u<U do10:     Update the particle velocity based on (22)11:     Update the particle position based on (23)12:  Perform crossover and mutation on particles position13:   Calculate the fitness function fit for the new particle position14:     $\mathrm{if}\;fit\left(X_{u}^{i+1}\right)<fit\left(P_{xbest}\right)$ then15:       
$P_{xbest}=X_{u}^{i+1}$
16:     end if17:     $\mathrm{if}\;fit\left(P_{xbest}\right)<fit\left(G_{xbest}\right)$ then18:       
$G_{xbest}=P_{xbest}$
19:     end if20:   end for21: end forOutput: Fitness function $fit\left(G_{xbest}\right)$


### 4.2. Algorithm Convergence Analysis

Rudolph et al. and Van et al. proved that the typical genetic algorithm and the BPSO algorithm are unable to converge to the global optima, respectively [29,30]. 

Solis et al. proposed the rules for the random search algorithm to converge to the global optima with probability 1 [31], stated as follows:Assumption (*H*1)
(24)$$f\left(D\left(x,\xi \right)\right)\leq f\left(x\right)\;and\;if\;\xi \in S,f\left(D\left(x,\xi \right)\right)\leq f\left(\xi \right),$$
where D is the function that generates the solution to the problem, ξ is the random vector generated from the probability space $\left(R^{n},B,\mu_{k}\right)$, f is the objective function, S is a subset of $R^{n}$, denotes the constraint space of the problem, $\mu_{k}$ is the probability measure on B, and B is the σ-domain of a subset of $R^{n}$.

Assumption (*H*2)
(25)$$For\;any\;\left(Borel\right)subset\;A\;of\;S\;with\;the\;measure\;v\left(A\right)\;>0,\;we\;have\;that\;\prod_{t=0}^{\infty }\left[1-\mu_{t}\left(A\right)\right]\;=\;0,$$
where $\mu_{t}\left(A\right)$ is the probability of generating A from the measure $v\left(A\right)$.

Convergence Theorem (Global Search)

Suppose that f is a measurable function, S is a measurable subset of $R^{n},$ and (H1) and (H2) are satisfied. Let ${\left\{x^{t}\right\}}_{t=0}^{\infty }$ be a sequence generated by the algorithm. Then,
(26)$$\underset{t\to \infty }{\lim}P\left[x^{t}\in R_{\varepsilon }\right]=1,$$
where $P\left[x^{t}\in R_{\varepsilon }\right]$ is the probability at step t, and $R_{\varepsilon }$ is the global best points set. The theorem shows that for any random search algorithm, it can converge to the global optimal with probability 1 as long as it satisfies Assumptions H1 and H2.

Next, we will analyze whether the GABPSO algorithm satisfies the above assumptions.

In the GABPSO algorithm, the solution sequence is $\left\{p_{g,t}\right\}$, where t is the number of evolutionary generations, and $p_{g,t}$ is the best particle position at the tth generation. The function D is defined as
(27)$$D\left(p_{g,t},x_{i}\left(t\right)\right)=\left\{\begin{array}{c} p_{g,t},f\left(p_{g,t}\right)\leq f\left(x_{i}\left(t\right)\right)\\ x_{i}\left(t\right),f\left(p_{g,t}\right)>f\left(x_{i}\left(t\right)\right) \end{array}\right..$$

Then, it is easy to prove that it satisfies Assumption H1.

To satisfy Assumption H2, the union of the sample space of a particle population of size N must contain S, i.e., $S\subseteq \overset{N}{\underset{i=1}{⋃}}M_{i,t}$, where $M_{i,t}$ is the support set for the sample space of ith particle at the tth generation. It has been shown that the basic PSO algorithm does not satisfy Assumption *H*2 [32,33]. In the PSO algorithm, as the number of iterations t increases, $v(M_{i,t})$ and $v(\overset{N}{\underset{i=1}{⋃}}M_{i,t})$ decrease. Thus, $v\left(\overset{N}{\underset{i=1}{⋃}}M_{i,t}\cap S\right)<v(S)$ is established, which means that there exists an integer $t^{\prime }$ such that when $t>t^{\prime }$, there exists a set A⊂S such that $\sum_{i=1}^{N}\mu_{i,t}(A)=0.$ This is not consistent with Assumption H2.

However, the GABPSO algorithm adds the crossover and mutation operations of the genetic algorithms. For a normally evolved particle, we set the union of its support set to α; for a particle recreated using crossover and mutation, we set the union of its support set to β. Due to the randomness and variability of crossover and mutation operations, there must exist an integer t2 such that β⊇S when t>t2. Therefore, for the GABPSO algorithm, there must exist an integer t2 such that α∪β⊇S when t>t2. Define any Borel subset of S to be $A=M_{i,t}$. When $v\left(A\right)>0,$ $\mu_{t}\left(A\right)=\sum_{i=1}^{N}\mu_{i,t}(A)=1,$ i.e., $\prod_{t=0}^{\infty }\left[1-\mu_{t}\left(A\right)\right]=0.$ Therefore, the GABPSO algorithm is satisfied by Assumption H2. 

According to the Convergence Theorem, it is known that the GABPSO algorithm can converge to the global optima with probability 1.

### 4.3. Algorithm Complexity Analysis

In the GABPSO algorithm, the population size is U, the number of iterations is I, and the problem size is N. For a single particle, the complexity of each operation is as follows:Velocity update: each particle gets a new velocity based on (25), and the time complexity is *O*(1)Position update: each particle gets a new position based on (26), and the time complexity is *O*(*N*).Fitness calculation: each particle needs to be calculated on the basis of the decision matrix *M* to obtain the corresponding fitness, and the time complexity is *O*(*N*).Fitness evaluation: each particle is compared with the historical best particle, and the time complexity is *O*(1)Crossover and mutation: each particle performs a crossover and mutation operation with a certain probability, and the time complexity is *O*(1)

Thus, for a single particle, the time complexity of one iteration is proportional to the problem size as *O*(*N*). Therefore, the time complexity of the algorithm is *IUO*(*N*)

### 4.4. Simulation Analysis

This research first created a 6 × 5 = 30 LEO satellite network based on the Iridium NEXT architecture. STK was used to obtain the shortest distance between satellites in each time slot in order to obtain the route table SatTEG. The specific scene parameters [19,21] and the related parameters of the GABPSO algorithm were set as shown in Table 2. In this paper, we provide the following reference algorithms for comparison to confirm the effectiveness of the proposed approach.

Binary particle swarm algorithm (BPSO): Guo et al. proposed a similar offloading problem in the satellite edge computing scenario and used the BPSO algorithm to provide a solution [21].Genetic algorithm (GA): Genetic algorithms are frequently utilized to resolve computational offloading issues in the ground cloud edge computing scenarios [34,35].Modified particle swarm algorithm (MPSO): Ren et al. proposed using the MPSO algorithm to solve the task-offloading problem of formation-flying satellites [19]. The authors inverted the flight velocity of some particles with a certain probability and perform position updates. The “variant particles” were created to obtain better search performance.

The same parameters were set equally throughout the algorithms to avoid losing generality, and the scene parameters were identical. Meanwhile, the relevant simulation results were averaged over 30 runs to reduce the impact of stochasticity.

Since the optimization objective of this study is the system cost obtained by weighting the delay and energy consumption, different combinations of weights will be studied first. The energy consumed during offloading is far greater than the time delay. When α is 1 and β is 0.1, the two are nearly equal. Therefore, we will keep α at 1 and explore the impact on the system cost as β grows. As illustrated in Figure 4, the energy consumption had an increasing impact on the system cost as β rose, and the system cost grew gradually. However, the GABPSO algorithm was able to solve for the lowest system cost whether the delay and energy consumption were essentially equal or the energy consumption was given much more importance than the delay. In subsequent simulations, α was set to 1 and β was set to 0.1 to simulate the scenarios where delay and energy consumption were equally important.

We first chose to explore the convergence performance of the four algorithms. It is noteworthy that we simulated two representative scenarios in which the high-load satellite ratios were 20% (30 × 20% = 6) and 60% (30 × 60% = 18), respectively. Most of the solutions in the search domain were defined as infeasible solutions when there was a high percentage of high-load satellites. When the number of high-load satellites is small, it will not have much impact. It was necessary to perform corresponding simulations to explore the possible consequences. The simulation figures demonstrate that the four algorithms performed nearly identically for two different conditions. In the process of particle evolution of the BPSO and MPSO algorithm, there were individual historical best position $P_{xbest}$ and the global best position $G_{xbest}$ of the particle population controlling the direction of the optimal solution. Therefore, compared with the simple GA, the BPSO and MPSO algorithms could move more quickly toward the optimal solution. These two algorithms, however, reached local optimality, while the particle swarm diversity vanished. The GABPSO algorithm combined the benefits of the PSO algorithm’s quick convergence and the GA algorithm’s robust search capacity, enabling speedy convergence to a better solution, as illustrated in Figure 5a. As the number of feasible solutions in the search domain decreased sharply, the convergence speed and optimal solution deteriorated. However, the GABPSO algorithm still outperformd the other three algorithms, which is shown in Figure 5b.

Firstly, we explored the relationship between system cost and the number of high-load satellites. The system cost of each algorithm gradually increased as the proportion of high-load satellites rose, as indicated in Figure 6. This is because there were more satellite nodes available that were closer to the source satellite node when there were fewer high-load satellites. This resulted in a smaller number of hops required for offloading through the ISL, which led to a subsequent decrease in transmission delay and energy consumption. In contrast, when the percentage of high-load satellites was large, there were fewer available satellite nodes closer to the source satellite node, causing inter-satellite offloading to be longer delayed and more energy-intensive. The GA algorithm was able to obtain lower system cost than the MPSO and BPSO algorithms. This was the same as the analysis of the convergence curve. The GABPSO algorithm, on the other hand, was able to maintain the best performance over time because it combined the advantages of both.

Next, we set the number of high-load satellites to 3 (30 × 10% = 3). On the basis of this, we studied the effect of other system parameters on the system cost. 

At first, we investigated how the quantity of original data in the subtasks would affect the system cost. As seen from Figure 7, the system cost grew gradually and with a more pronounced trend. The increased quantity of original data for subtasks led directly not only to an increase in computational delay and energy consumption but also to an increase in inter-satellite transmission delay and energy consumption. It led to a relatively fast curve change. Similarly, the GABPSO algorithm consistently outperformed the other baseline algorithms. The MPSO algorithm used the inverse of the particle velocity to achieve the particle variation. It was difficult to make essential changes to the particle population, and the entire search domain could not be fully explored. As a result, the MPSO method did not significantly outperform the BPSO algorithm.

Further, we explored the changes brought about by computational complexity. The increase in computational complexity indicated that the task took longer to compute on the satellite nodes. This, in turn, caused an increase in the overall system cost. As depicted in Figure 8, all four algorithm times increased as the computational complexity grew. The MPSO algorithm and BPSO algorithm lacked powerful global search capability for a better solution, leading to the worst performance, while the GABPSO algorithm performed the best.

Finally, we incrementally increased ISL bandwidth while holding the other variables constant to investigate how the transmission capacity of ISL affects the system cost. Figure 9 shows how effectively the ISL bandwidth affected the system cost. The increased bandwidth enabled faster offloading of subtasks among satellite nodes. It allowed for the system cost to be gradually reduced as well, and the GABPSO algorithm still performed the best. The effect of the same growth on system cost was more noticeable when the ISL bandwidth was minimal. When the ISL bandwidth is large enough, the inter-satellite transmission delay and energy consumption will be negligible. In that case, the system cost will almost equal the delay and energy consumption required for the computation. It is anticipated that inter-satellite task scheduling will be able to be finished in a relatively short time in the future if satellite transmission performance is markedly enhanced.

In the above simulation scenarios, the GABPSO algorithm was always able to achieve the best performance because it took into account the fast converge capability of the BPSO algorithm and the variational properties of the GA. Due to its cross-mutation property, the GA algorithm was also better able to look for better solutions. The MPSO and BPSO algorithms had the worst overall performances because they tended to fall into local optima. The simulation results are consistent with the analysis in the convergence curve.

### 4.5. Statistical Analysis

The simulation findings provided some support for the GABPSO algorithm’s superiority. Referring to research [36], a two-way analysis of variance (ANOVA) was used to explore the system cost in relation to each parameter for a more in-depth analysis. This process was used to test the effect of two factors on the dependent variable, consistent with the type of simulation in this study. Firstly, the objective was defined to test whether there was any difference in the scheduling algorithms or high-load satellite proportions at the 0.05 level of standard significance. The calculation parameters are shown in Table 3.

Step 1: Null Hypotheses:H0^(1)^: There is no significant difference in the scheduling algorithmsH0^(2)^: There is no significant difference in the high-load satellite proportions Alternative Hypotheses:H1^(1)^: There is a significant difference in the scheduling algorithmsH1^(2)^: There is a significant difference in the high-load satellite proportionsStep 2: In this scenario, *a* = 4 and *b* = 7. At the 0.05 level of significanceH0^(1)^: $F_{\alpha }((a-1),(a-1)(b-1))=F_{0.05}(3,18)=3.16$H0^(2)^: $F_{\alpha }((b-1),(a-1)(b-1))=F_{0.05}(6,18)=2.66$Step 3: CalculationTotal sum of squares: $SST=\sum_{i=1}^{a}\sum_{j=1}^{b}{\left(x_{ij}-\overset{=}{x}\right)}^{2}$ = 667,536.11Variation between rows: $SSR=\sum_{i=1}^{a}\sum_{j=1}^{b}{\left({\overset{-}{x}}_{i.}-\overset{=}{x}\right)}^{2}$ = 195,397.82Variation between columns: $SSC=\sum_{i=1}^{a}\sum_{j=1}^{b}{\left({\overset{-}{x}}_{.j}-\overset{=}{x}\right)}^{2}$ = 461,881.36Variation due to error: SSE=SST−SSR−SSC = 10,256.93The specific results of the two-way ANOVA analysis are shown in Table 3.Step 4: DecisionAs $\left[Fr=114.30\right]>\left[F_{\alpha }\left(\left(a-1\right),\left(a-1\right)\left(b-1\right)\right)=3.16\right]$, we can reject the $H0^{\left(1\right)}$ at the 0.05 level of significance. The results of the analysis show that different scheduling algorithms produced significant differences. In addition, as $\left[Fc=135.09\right]>\left[F_{\alpha }\left(\left(b-1\right),\left(a-1\right)\left(b-1\right)\right)=2.66\right]$, this verifies that different proportions of high-load satellites can also have a significant impact. Similarly, other simulation results were analyzed, and the specific results are shown in Table 4.

## 5. Conclusions

The proposed research work explores the problem of offloading a single dependent task to multiple satellites for collaborative processing in the satellite edge computing scenario. Firstly, a model is proposed in which tasks are offloaded to multiple satellites for collaborative computing without the participation of high-load satellites. Secondly, the crossover and mutation operations of the GA are introduced in this paper to address the drawbacks of the traditional BPSO algorithm. Utilizing the optimized GABPSO algorithm, a lower system cost can be obtained under the scenario. The experiments verified that the optimized algorithm has better performance than other baseline algorithms. 

In practical satellite applications, the size of remote sensing images is huge. Adopting the strategy proposed in this paper can effectively accelerate the on-orbit analysis of images. In addition, in the future satellite-IoT architecture, the analysis of ground monitoring data acquired by satellites, etc., is also applicable to the scenario studied in this work. The strategy proposed in this paper has some reference value for all these applications.

## 6. Future Works

The scenarios studied in this work address only the single-user, single-service scenario. Future research will concentrate on the task-scheduling problem of the multi-user, multi-service satellite scenario. Furthermore, in practical engineering, the arrival of tasks is continuous. Dynamic scheduling for continuous tasks also requires further research.

In terms of the algorithm, optimization objectives and heuristic algorithms are combined through node mapping and algorithmic coding approaches in this study. Using the algorithm’s own search to find the optimal solution will consume more time, and combining some a priori methods can effectively improve the search velocity. In addition, the decision matrix and the criteria defined in this paper will take up a lot of space. If these problems are sufficiently improved in future work, they will have high engineering value.

## Figures and Tables

**Figure 1 sensors-23-04271-f001:**
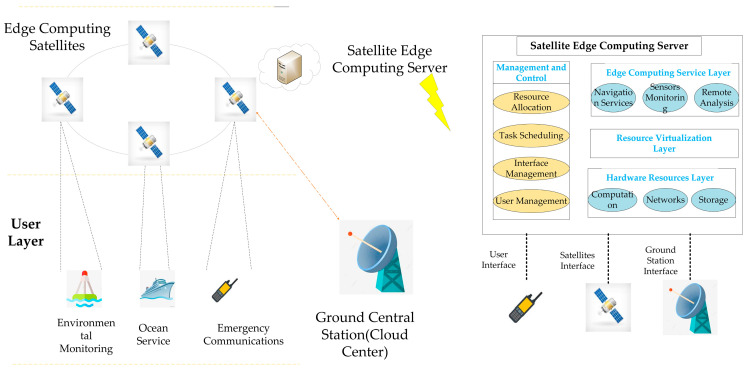
Satellite edge computing architecture.

**Figure 2 sensors-23-04271-f002:**
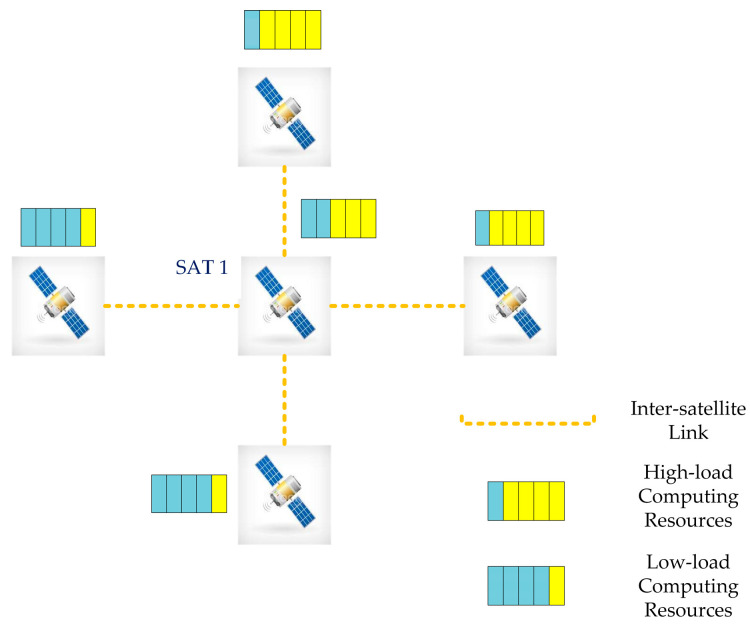
Satellite Communication Model.

**Figure 3 sensors-23-04271-f003:**
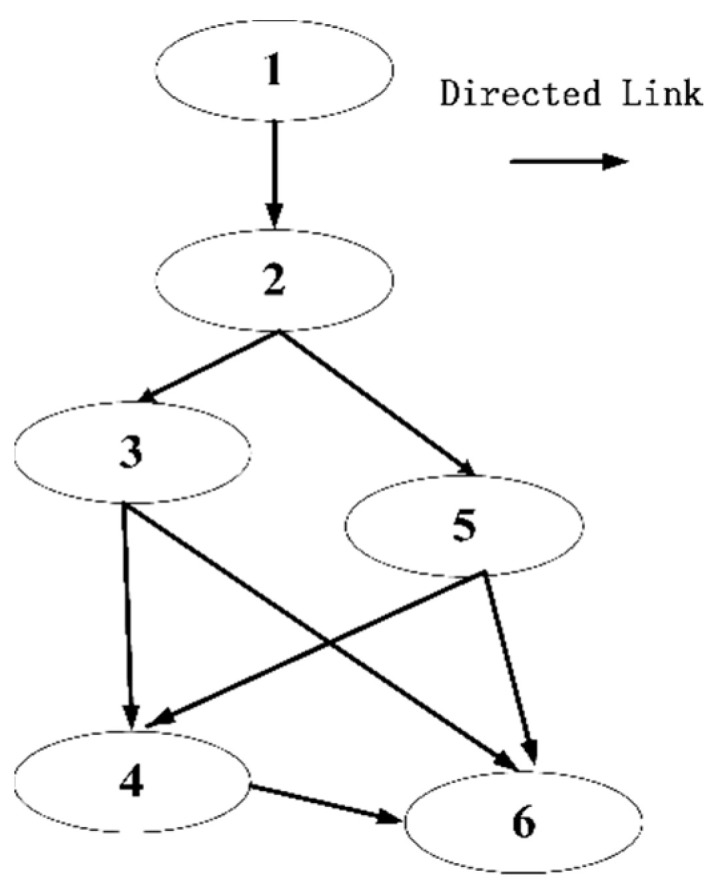
DAG model.

**Figure 4 sensors-23-04271-f004:**
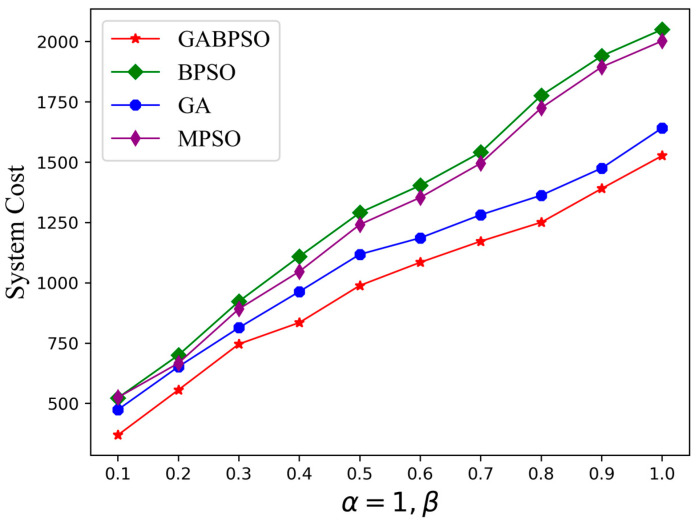
System cost vs. the value of β.

**Figure 5 sensors-23-04271-f005:**
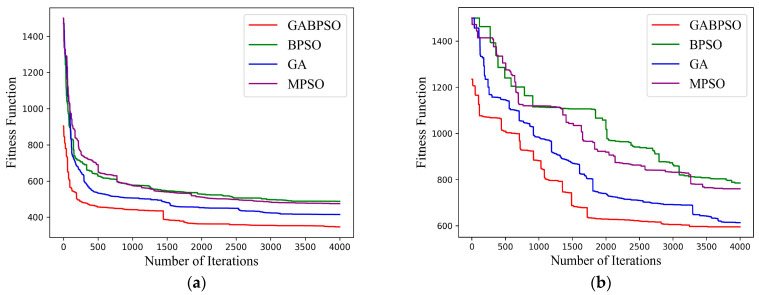
(**a**) Convergence curve for 6 high-load satellites; (**b**) Convergence curve for 18 high-load satellites.

**Figure 6 sensors-23-04271-f006:**
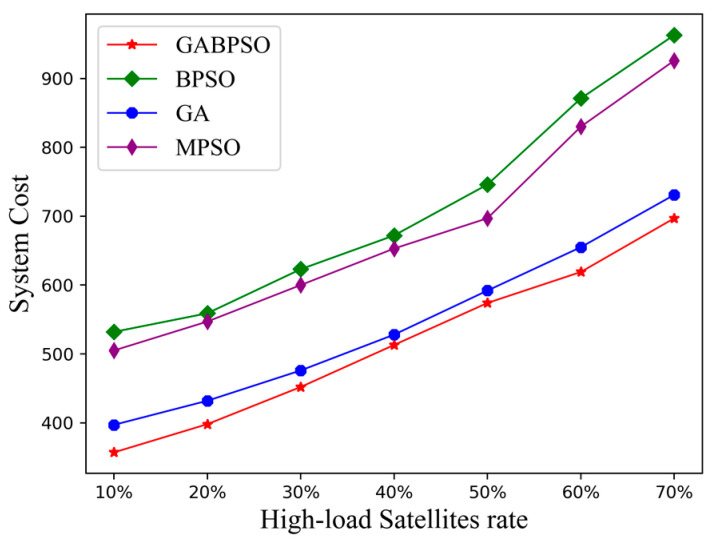
System cost vs. high-load satellite rate.

**Figure 7 sensors-23-04271-f007:**
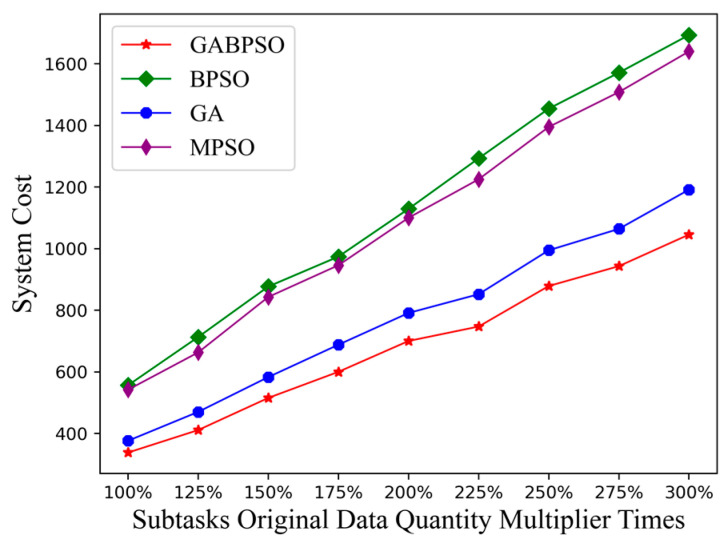
System cost vs. subtasks original data quantity.

**Figure 8 sensors-23-04271-f008:**
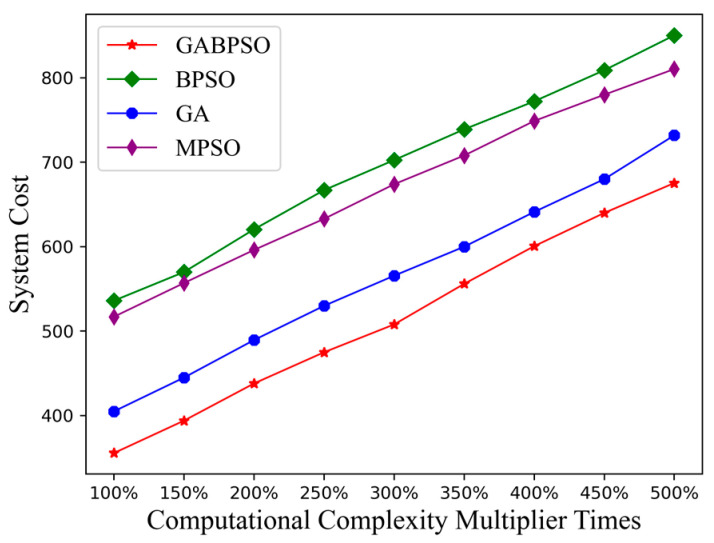
System cost vs. computational complexity.

**Figure 9 sensors-23-04271-f009:**
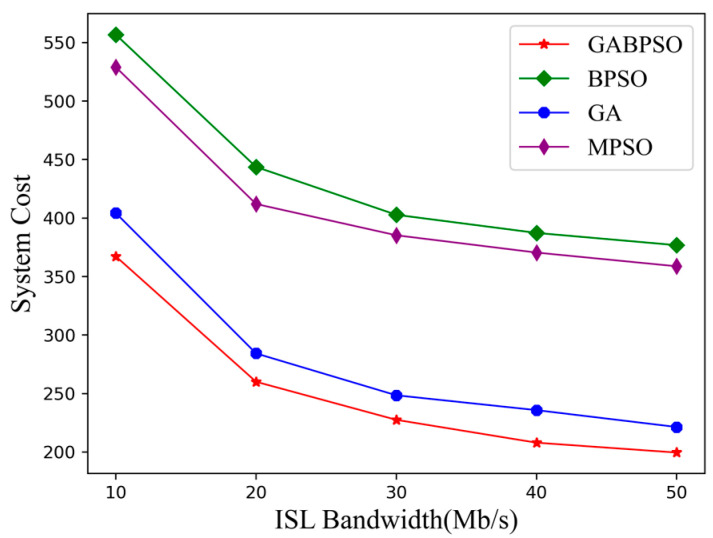
System cost vs. ISL bandwidth.

**Table 1 sensors-23-04271-t001:** A summary of the literature.

References	Independent Task	Dependent Task	Dynamic Network Topologies	High-Load Satellites	Task Delay	Energy Consumption
[11,19]	√				√	
[20]		√			√	
[4,21]		√	√		√	
Our paper		√	√	√	√	√

**Table 2 sensors-23-04271-t002:** Parameter setting.

**Algorithm Parameter**	**Value**
Maximum number of iterations, I	4000
Number of particles, U	100
Learning factor, C1,C2	2, 2
Inertia weights, W	0.8
$\mathrm{Crossover}\;\mathrm{probability},\;P_{c}$	0.6
$\mathrm{Mutation}\;\mathrm{probability},\;P_{m}$	0.1
**Scene Parameter**	**Value**
Time slot,Δt	10 s
Subtask original data size, D	[200, 250] Mb
Subtask result data size, RD	[2, 5] Mb
Computational complexity, ζ	$\frac{1900}{8}$
ISL bandwidth, B $\mathrm{ISL}\;\mathrm{transmission}\;\mathrm{power},\;P_{trans}$ $\mathrm{Satellite}\;\mathrm{computational}\;\mathrm{power},\;P_{com}$	10 Mb/s 100 J/s [5, 10] J/s
$\mathrm{Satellite}\;\mathrm{computational}\;\mathrm{performance},\;C_{k}$	[5, 10] Ghz

**Table 3 sensors-23-04271-t003:** Calculation parameters.

Scheduling Algorithms	High-Load Satellite Proportions as 10%	High-Load Satellite Proportions as 20%	High-Load Satellite Proportions as 30%	High-Load Satellite Proportions as 40%	High-Load Satellite Proportions as 50%	High-Load Satellite Proportions as 60%	High-Load Satellite Proportions as 70%
BPSO	532.41	559.63	623.39	672.36	746.15	871.16	963.73
GA	397.12	432.65	476.35	528.73	592.66	655.53	731.26
GABPSO	357.77	398.24	452.85	513.35	574.75	619.83	697.98
MPSO	505.32	547.28	600.11	653.75	692.68	830.77	926.85

**Table 4 sensors-23-04271-t004:** ANOVA results.

Sources	Variance	Degrees of Freedom	Mean Square	F-Value
Algorithms	SSR= 125,633.16	$\left(a-1\right)=3$	$\Delta S_{R}^{2}=\frac{SSR}{a-1}=$ 41,877.72	$Fr=\frac{\Delta S_{R}^{2}}{\Delta S_{E}^{2}}$ = 2765.48
ISL bandwidth	SSC = 80,806.03	$\left(b-1\right)=4$	$\Delta S_{C}^{2}=\frac{SSC}{b-1}=$ 20,201.51	$Fc=\frac{\Delta S_{C}^{2}}{\Delta S_{E}^{2}}$ = 1334.05
Error	SSE= 181.72	$\left(a-1\right)\left(b-1\right)=12$	$\Delta S_{E}^{2}=\frac{SSE}{\left(a-1)(b-1\right)}=$ 15.14	-
Total	SST = 206,620.90	$\left(ab-1\right)=19$	-	-
Algorithms	SSR= 196,415.00	$\left(a-1\right)=3$	$\Delta S_{R}^{2}=\frac{SSR}{a-1}=$ 65,471.67	$Fr=\frac{\Delta S_{R}^{2}}{\Delta S_{E}^{2}}$ = 1759.82
Computational complexity	SSC = 366,858.53	$\left(b-1\right)=8$	$\Delta S_{C}^{2}=\frac{SSC}{b-1}=$ 45,857.32	$Fc=\frac{\Delta S_{C}^{2}}{\Delta S_{E}^{2}}$ = 1232.60
Error	SSE= 892.89	$\left(a-1\right)\left(b-1\right)=24$	$\Delta S_{E}^{2}=\frac{SSE}{\left(a-1)(b-1\right)}=$ 37.20	-
Total	SST = 564,166.42	$\left(ab-1\right)=35$	-	-
Algorithms	SSR= 1,383,009.98	$\left(a-1\right)=3$	$\Delta S_{R}^{2}=\frac{SSR}{a-1}=$ 461,003.33	$Fr=\frac{\Delta S_{R}^{2}}{\Delta S_{E}^{2}}$ = 80.08
Subtasks original data quantity	SSC = 3,325,662.57	$\left(b-1\right)=8$	$\Delta S_{C}^{2}=\frac{SSC}{b-1}=$ 415,707.82	$Fc=\frac{\Delta S_{C}^{2}}{\Delta S_{E}^{2}}$ = 72.21
Error	SSE= 138,166.68	$\left(a-1\right)\left(b-1\right)=24$	$\Delta S_{E}^{2}=\frac{SSE}{\left(a-1)(b-1\right)}=$ 5756.94	-
Total	SST = 4,846,839.23	$\left(ab-1\right)=35$	-	-
Algorithms	SSR= 742,694.99	$\left(a-1\right)=3$	$\Delta S_{R}^{2}=\frac{SSR}{a-1}=$ 247,565.00	$Fr=\frac{\Delta S_{R}^{2}}{\Delta S_{E}^{2}}$ = 32.51
β	SSC = 6,956,029.66	$\left(b-1\right)=9$	$\Delta S_{C}^{2}=\frac{SSC}{b-1}=$ 772,892.18	$Fc=\frac{\Delta S_{C}^{2}}{\Delta S_{E}^{2}}$ = 101.5
Error	SSE= 205,587.88	$\left(a-1\right)\left(b-1\right)=27$	$\Delta S_{E}^{2}=\frac{SSE}{\left(a-1)(b-1\right)}=$ 7614.37	-
Total	SST = 7,904,312.53	$\left(ab-1\right)=39$	-	-

## Data Availability

No new data were created or analyzed in this study. Data sharing is not applicable to this article.

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
