# Peer review of "An On-Orbit Task-Offloading Strategy Based on Satellite Edge Computing"

_sensors, 2023, doi:10.3390/s23094271_

Round 1

Reviewer 1 Report

This article explores the problem of offloading a single dependent task to multiple satellites for collaborative processing in the satellite edge computing scenario. Some of the comments that could improve the quality of this article are listed below.

1.      Contributions should be clearly summarized, as the current version is fragmented.

2.      Edge task scheduling have been studied in many current works. How is satellite task scheduling different from traditional edge task scheduling? How does this distinction manifest itself in this paper?

3.      Some abbreviations have the first letter in lowercase, some are in uppercase, and some are not even explained. For example, QoS, etc.

4.      The format of references needs to be standardized.

5.      There are some grammatical errors in the manuscript. The authors should carefully proofread the whole manuscript to correct such errors.

Author Response

Comment #1.      Contributions should be clearly summarized, as the current version is fragmented.

Response: Thanks for your valuable suggestion. We have rewritten the Contributions to describe the scenarios studied and the algorithms applied in this paper, respectively. The innovations of this paper are also mentioned. In addition, a new table describing the differences between the work in this paper and the existing work has been added. The modified parts are marked in blue font.

Comment #2.      Edge task scheduling have been studied in many current works. How is satellite task scheduling different from traditional edge task scheduling? How does this distinction manifest itself in this paper?

Response: Thanks for your valuable suggestion. The biggest difference between satellite task scheduling and traditional task scheduling is that satellite nodes make periodic movements around the Earth. This makes the routing path between satellites changes with each scheduling time slot, resulting in varying latencies for data transmission. To address this issue, we first constructed the ISL building rule  in Section 3.2, which enables us to determine the transmission hops for tasks at all scheduling time slots. In Section 3.5, the transmission delay of a task is directly related to the transmission hops, which has a significant impact on the task completion delay. Meanwhile, we explore the impact of the bandwidth of the ISL and the amount of task data on the optimization objective in our simulations. Their variation then also affects the transmission delay.

Comment #3.      Some abbreviations have the first letter in lowercase, some are in uppercase, and some are not even explained. For example, QoS, etc.

Response: Thanks for your valuable suggestion. We carefully checked the full text and added the original text of all abbreviations. And the correct abbreviation format is also ensured. The modified parts are marked in blue font.

Comment #4.      The format of references needs to be standardized.

Response: Thanks for your valuable suggestion. We carefully checked the format of the references and found that the format of the conference paper was not in compliance with the requirements. We modified them according to the MDPI format and marked them in blue font.

Comment #5.      There are some grammatical errors in the manuscript. The authors should carefully proofread the whole manuscript to correct such errors.

Response: Thanks for your valuable suggestion. We carefully checked the manuscript and corrected any grammatical errors and spelling mistakes in it.

Reviewer 2 Report

This work focuses on task offloading to balance computational latency and energy consumption on a satellite edge computing infrastructure. The research topic is exciting and timely. However, there are some significant concerns about the manuscript:

-Abstract

-The abstract could be more specific in describing the research contribution. It currently provides a general overview of the paper, but it would be more helpful to highlight the particular findings of the study.

-Section 1

-The authors could provide more context for the TIANSUAN Constellation test satellite and the validation it provides for remote sensing image inference and computing services.

-Section 2

-This section could benefit from a more critical analysis of the existing research. For example, rather than simply summarizing what other authors have done, evaluating their approaches' strengths and weaknesses would be helpful and identifying areas where further research is needed.

-This section should also be more precise about what the current paper contributes to the field. For example, while the authors state that they are focusing on static task scheduling, it needs to be clarified what specific contributions they are making or how their approach differs from existing work.

-Section 3

-In (18), how are alpha and beta weights determined? What are the factors influencing these weights?

-The objective function in (19) should mention the decision variable M.

-Section 4

-The proposed metaheuristic algorithm GABPSO gives the approximate solution of the optimization problem defined in (19). The authors should mention why finding the exact solution to the optimization problem is infeasible. 

-Please mention the inputs to Algorithm 1.

-The values of simulation parameters hop and C are not mentioned in Table 2.

-Section 5

-This section could benefit from a more detailed discussion of the implications and significance of the findings. For example, the authors briefly mention the proposed strategy for multi-satellite collaborative processing tasks. Still, providing more context and examples to support this claim would be helpful.

-Section 6

-This section needs to be more cohesive, with several ideas presented without clear organization or prioritization. In addition, it would be beneficial to provide a more structured discussion of future research directions, grouping similar ideas or prioritizing the most critical areas for future investigation.

-The phrase "using reinforcement learning to solve dynamic scheduling under continuous task generation deserves further study" could be made more specific by identifying the particular aspects of the problem the authors intend to investigate.

-The writing could be improved in some places, with more attention to grammar and syntax. For example, in several places, the authors use awkward phrasing or sentence structures that make it difficult to follow their meaning.

Round 2

Reviewer 2 Report

The authors have addressed most of the comments. However, the following two critical comments remain unaddressed:

-In (18), how are alpha and beta weights determined? What are the factors influencing these weights? This is needed for replicating the proposed scheme in real systems.

-The proposed metaheuristic algorithm GABPSO gives the approximate solution of the optimization problem defined in (19). The authors should mention why finding the exact solution to the optimization problem is infeasible. This is to justify why we cannot directly use solutions to the problem (19) generated by traditional solvers such as Gurobi, CPLEX, CVX, etc.

I have one additional comment:

-Page 4, Line 146: Reference error.

Author Response

Comment #1-In (18), how are alpha and beta weights determined? What are the factors influencing these weights? This is needed for replicating the proposed scheme in real systems.

Response: Thank you for the important comment. In the existing literature, most studies have focused only on minimizing the task completion delay. Indeed, latency is a metric that is of interest to all users. However, the satellite is powered by solar energy. When it works in the shadow part of the Earth, its energy resources are very precious. So how to minimize the energy consumption from mission scheduling is also a valuable optimization goal. This paper provides some reference for this problem. But we cannot give specific alpha and beta because it has different latency tolerance and energy consumption tolerance for different types of tasks. In this paper, we vary the value of beta to imply the change of algorithm performance under different latency and energy consumption concerns. This provides a reference value for future work. If subsequent researchers investigate strategies for specific types of tasks, more precise alpha and beta should be provided. Also, the application of the algorithm in this paper to real systems requires further exploration of specific types of tasks. This work still needs to be developed in detail in the future.

Comment #2-The proposed metaheuristic algorithm GABPSO gives the approximate solution of the optimization problem defined in (19). The authors should mention why finding the exact solution to the optimization problem is infeasible. This is to justify why we cannot directly use solutions to the problem (19) generated by traditional solvers such as Gurobi, CPLEX, CVX, etc.

Response: Thank you for the important comment. The GABPSO algorithm proposed in this paper is theoretically capable of solving for the exact optimal solution. However, the number of iterations and parameter settings make the algorithm may only search for the approximate optimal solution. In fact, the optimization objective of this paper is a binary optimization-like problem. The search space is a finite set of discrete [0, 1]. Such a problem is capable of being solved to the exact solution. Therefore, solvers such as CPLEX and CVX should be able to solve the problem. The motivation for this paper comes from the analysis of the defects of the BPSO algorithm used in the literature 21. So we improve the algorithm on this basis. We actually do not have the knowledge of solvers such as CPLEX, CVX, etc., and have not tried to solve with them before. Related studies can be developed subsequently. Note that if solving the exact solution by solver is a very time consuming option, then we do not recommend it. If it is to be applied to a real system, it is acceptable to find an approximate optimal solution in finite time. This is just a simple hypothesis for us, the differences would require more simulations and comparisons to distinguish, which could be organized into a new paper.

Comment #3-Page 4, Line 146: Reference error.

Response: Thank you for the important comment. In Table 1, we collated the studies from the existing literatures. We found errors in the summary of some of the studies. For the optimization objectives of these articles, they can be divided into minimizing the delay and minimizing the energy consumption. Some articles will have some new definitions for their optimization objectives, but they are still related to latency. We categorize them as minimizing delay. We incorrectly counted literature 11 and literature 19 as minimizing energy consumption. We rechecked and found this problem and modified it.